# Effectiveness of Repeated Stereotactic Body Radiation Therapy for Hepatocellular Carcinoma—Consideration of the Locations of Target Lesions

**DOI:** 10.3390/cancers15030846

**Published:** 2023-01-30

**Authors:** Shigeki Yano, Tomoki Kimura, Tomokazu Kawaoka, Takahiro Kinami, Shintaro Yamasaki, Yusuke Johira, Masanari Kosaka, Kei Amioka, Kensuke Naruto, Yuwa Ando, Kenji Yamaoka, Yasutoshi Fujii, Shinsuke Uchikawa, Hatsue Fujino, Atsushi Ono, Takashi Nakahara, Eisuke Murakami, Wataru Okamoto, Masami Yamauchi, Michio Imamura, Junichi Hirokawa, Yasushi Nagata, Hiroshi Aikata, Shiro Oka

**Affiliations:** 1Department of Gastroenterology, Graduate School of Biomedical and Health Sciences, Hiroshima University, Hiroshima 734-8551, Japan; 2Department of Radiation Oncology, Kochi University, Kochi 783-8505, Japan; 3Department of Radiation Oncology, Hiroshima University, Hiroshima 734-8551, Japan; 4Department of Gastroenterology, Hiroshima Prefectural Hospital, Hiroshima 734-0004, Japan

**Keywords:** hepatocellular carcinoma, SBRT, repeated, local therapy, local recurrence, overall survival

## Abstract

**Simple Summary:**

Many treatment options are available for hepatocellular carcinoma. Radiation therapy is one of the local options, along with surgery and radiofrequency ablation. Stereotactic body radiation therapy is an effective therapy for hepatocellular carcinoma that provides good outcomes with tolerable toxicities and has been recognized as a local therapy for patients who are ineligible for surgery or radiofrequency ablation. Hepatocellular carcinoma tends to recur, so stereotactic body radiation therapy is also used for recurrent cases. However, few reports have summarized local tumor control, transition of liver function, and side effects in cases of repeated stereotactic body radiation therapy. This study retrospectively evaluated the efficacy of stereotactic body radiation therapy, including repeated radiation therapy, for hepatocellular carcinoma.

**Abstract:**

The present study retrospectively evaluated the efficacy of stereotactic body radiation therapy (SBRT), including repeated SBRT, for hepatocellular carcinoma. Participants comprised 220 HCC patients treated with SBRT in Hiroshima University Hospital between December 2008 and December 2021. Median overall survival (OS) and disease-free survival were 52 months (range, 45–64 months) and 17 months (range, 14–23 months), respectively. The 5-year local tumor recurrence rate was 3.4% (95% confidence interval (CI), 1.3–6.9%). Fifty-three patients underwent repeated SBRT (twice, 53 cases; three times, 10 cases; four times, 4 cases; five times, 1 case). Median interval between first and second SBRT was 20 months. Median OS from first SBRT was 76 months (95% CI, 50–102 months). Among patients with repeated SBRT, only one case showed local recurrence after second SBRT. Albumin–bilirubin score increased significantly from 6 to 12 months after repeated SBRT, both in the same segment and in remote segments, but the increase was not significant in the same segment. Only one case of grade 3 bile duct stricture was observed in patients who were treated with repeated SBRT. In conclusion, repeated SBRT provides good local control and a low risk of side effects.

## 1. Introduction

Primary liver cancer was the sixth most diagnosed cancer and third leading cause of cancer death worldwide in 2020 [1]. Most primary liver cancers are hepatocellular carcinoma (HCC), comprising 75–85% of cases [1]. In Japan, the treatment strategy for HCC is based on the JSH HCC Guidelines 2021 [2], published by the Japan Society of Hepatology. The treatment algorithm is determined by liver function, metastasis, vascular invasion, tumor number, and tumor size. Early-stage HCC warrants radical cure by local treatment. In addition to curative treatments for early-stage HCC, including liver transplantation, surgical resection, and radiofrequency ablation (RFA), stereotactic body radiation therapy (SBRT) also achieves satisfactory local control rates in HCC patients, and provides another option for patients. Numerous studies have identified SBRT as an effective method for patients with different stages and sizes of HCC [3,4,5,6,7,8,9,10,11,12,13].

SBRT can be performed even for patients for whom surgery or RFA is unsuitable due to age, complications, the target lesion is close to vessels, bile ducts, or the diaphragm, or difficulty because the lesion cannot be visualized by ultrasound [14]. On the other hand, the recurrence rate of HCC after surgery [15] and RFA [16] has been reported to reach 70–80% at 5 years, and few reports have described the use of repeated SBRT as a treatment for recurrence. In addition, many issues remain unknown regarding changes in liver function and side effects after repeated SBRT.

In this report, we retrospectively evaluated not only local tumor control and survival following SBRT for HCC, but also those for repeated SBRT.

## 2. Materials and Methods

### 2.1. Patients

A total of 223 patients received SBRT for HCC between December 2008 and December 2021 at Hiroshima University Hospital. We selected patients who met the following criteria: (i) unsuitable for surgical resection or radiofrequency ablation; (ii) ≤3 nodules, each ≤5 cm in a diameter; (iii) no metastasis; and (iv) Child–Pugh scores ≤7. Three patients were excluded because their Child–Pugh scores were >8, but 220 patients met the criteria. Among these, 167 patients were treated with SBRT only once. The remaining 53 patients were treated with SBRT twice or more (twice, 53 cases; three times, 10 cases; four times, 4 cases; five times, 1 case). Cases with two or more prior radiotherapy treatments were considered as cases of repeated SBRT. Among these, we divided patients into those whose lesions were in the same segment (e.g., first lesion in S7, and recurrent lesion not as local recurrence (LR), but occurring within the same S7 zone) and those whose lesions were in a remote segment according to computed tomography (CT) before SBRT. Among cases of repeated SBRT, 13 cases received irradiation in the same segment and 40 cases received irradiation in a remote segment.

### 2.2. Therapeutic Method

The SBRT treatment procedure was taken from published reports [3,12,13]. In summary, SBRT was performed as three-dimensional conformal radiation therapy or volumetric modulated arc therapy. When SBRT for HCC was first introduced in Hiroshima University Hospital, isocenter prescription was used as a method of prescribing based on the dose received by the tumor center. The protocol was started by dividing the dose into 60 Gy in 8 fractions to the central region and 48 Gy in 4 fractions to peripheral. However, since 2014, the protocol has been changed to the planning target volume (PTV) D95% prescription, which is a method of prescribing based on the radiation dose received by the entire tumor. The dose of 48 Gy in 4 fractions, which was prescribed at the isocenter, was almost equal to 40 Gy in 4 fractions, covering at least 95% of the PTV (D95%) at the 80% isodose line. All cases were standardized to 40 Gy in 4 fractions.

From the second SBRT, although the deformable registration method was used, the previously irradiated area was not considered in each treatment plan because identifying the exact location and area of the tumor and adjacent highly irradiated liver tissue was difficult due to local atrophy from the previous SBRT. Combination with transarterial chemoembolization (TACE), which involved lipiodol with chemotherapeutic agents, was allowed before SBRT. The median interval between TACE and SBRT was 1 month.

### 2.3. Follow-Up after Treatment

Follow-up was defined from the start date of SBRT, and patients underwent laboratory tests and dynamic CT 3, 6, and 12 months after completing SBRT. Laboratory tests included total bilirubin, albumin, and prothrombin time to calculate the Child–Pugh score. The albumin–bilirubin (ALBI) score was also evaluated to assess liver function simply [17]. The ALBI score is directly calculated based on serum bilirubin and albumin values in the clinical setting using the following formula: ALBI score = (log_10_ bilirubin × 0.66) + (albumin × −0.085), where bilirubin is in micromoles per liter and albumin is in grams per liter. A lower ALBI score suggests better liver function. ALBI score is classified into three grades: Grade 1, score ≤−2.60; Grade 2, score >−2.60 but ≤−1.39; and Grade 3, score >−1.39). ALBI grade is reportedly more useful than the Child–Pugh score for predicting outcomes (i.e., assessing liver function) of systemic drug therapy, surgery, and RFA because it is much simpler and relying on fewer variables [17].

Tumor responses were assessed according to the modified response evaluation criteria in solid tumors (mRECIST), with assessment of tumor necrosis based on nonenhanced areas [18]. Local tumor progression was defined as progressive disease according to mRECIST, and LR was defined as local progression within the PTV. Newly developed tumors outside the PTV were considered intrahepatic recurrences.

Biochemical and hematological toxicities (i.e., abnormalities) in total bilirubin, alanine aminotransferase, aspartate aminotransferase, alkaline phosphatase, platelet count, (and albumin) and bile duct stricture, portal vein thrombosis, ascites, pneumonia, and gastrointestinal disorders were also evaluated after SBRT according to the Common Terminology Criteria for Adverse Events (version 5.0).

Radiation-induced liver disease (RILD) as a specific form of radiotherapy-induced liver damage and the presence of hepatomegaly, nonmalignant ascites, or elevated alkaline phosphatase within 3 months after the start of radiotherapy was also evaluated.

### 2.4. Statistics

The Kaplan–Meier method was used to calculate overall survival (OS) and disease-free survival (DFS). OS and DFS were calculated from the starting date of SBRT until the date of final follow-up or death and the date of recurrence, respectively.

LR rate was calculated using the Fine and Gray test for cases of recurrence within the PTV within 5 years after irradiation, with death before LR as a competing risk. The Mann–Whitney U test and Fisher’s exact test were used to compare background characteristics of patients. The Friedman test and Bonferroni’s multiple comparisons were used to compare ALBI scores.

All statistical analyses were performed using EZR (Saitama Medical Center, Jichi Medical University), which is a graphical user interface for R (The R Foundation Computing, version 3.4.1). Values of *p* < 0.05 were defined as statistically significant.

## 3. Results

### 3.1. Patient Characteristics

The flow chart of this study is shown in Figure 1. Between 2008 and 2021, a total of 223 patients with HCC with 321 tumors were treated using SBRT at Hiroshima University Hospital. Among these, three patients were excluded because their Child–Pugh scores were ≥8. The remaining 220 patients with 318 tumors who fulfilled the inclusion criteria were enrolled. Of these, 167 patients were treated with SBRT only once. Of the 53 patients treated with repeated SBRT, 13 patients were irradiated in the same segment and 40 patients were irradiated in a remote segment.

The background characteristics of patients are shown in Table 1. Median age was 76 years (range, 38–95 years). The etiology of liver cirrhosis was hepatitis B virus, hepatitis C virus, and non-B, non-C viral hepatitis in 32 (14.5%), 143 (65%), and 48 (21.8%) patients, respectively. Child–Pugh scores were 5, 6, and 7 in 148 (67.3%), 49 (22.3%), and 23 patients (10.4%), respectively. Median tumor size was 16 mm (range, 8–50 mm). The 48 patients (21.8%) treated with first SBRT as the initial treatment for HCC were regarded as naïve cases. The remaining 172 patients (78.2%) received first SBRT for LR or intrahepatic recurrent HCC after other treatment modalities.

Fifty-three cases were treated with SBRT twice or more (repeated SBRT). Median age was 76 years (range, 54–89 years). The etiology of liver cirrhosis was hepatitis B virus, hepatitis C virus, non-B-non-C viral hepatitis in 7 (13.2%), 39 (73.6%), and 7 patients (13.2%), respectively. Median tumor size in all SBRT courses was 17 mm (range, 8–35 mm). Four patients (7.5%) were treated with first SBRT as initial treatment for HCC.

### 3.2. Treatment Outcomes

Figure 2 shows OS for total cases treated with SBRT and repeated cases treated with SBRT. Median survival time (MST) for total cases was 52 months (95% confidence interval (CI), 45–64 months). Three- and 5-year OS rates were 65.3% (95% CI, 58.1–71.6%) and 45.6% (95% CI, 37.6–53.1%), respectively. Regarding repeated SBRT, MST from first SBRT was 76 months (95% CI, 50–102 months). Three- and 5-year OS rates for repeated cases were 78.2% (95% CI, 64.0–87.3%) and 61.5% (95% CI, 45.8–73.9%), respectively.

Figure 3 shows the DFS of total cases and repeated cases. Median DFS of total cases was 17 months (95% CI, 14–23 months). For repeated cases, median DFS from first SBRT was 13 months (95% CI, 9–17 months).

Figure 4 shows the LR rate. LR was identified in six patients in total, including one case after second SBRT. Three- and 5-year LR rates were 2.5% (95% CI, 0.4–3.8%) and 3.4% (95% CI, 1.3–6.9%), respectively. Regarding repeated SBRT, only one case showed LR within 1 year after second SBRT.

### 3.3. Differences between Same-Segment and Remote-Segment Cases

Among the cases with repeated SBRT, we examined differences between patients whose lesions were in the same segment and those with lesions in a remote segment. Of the 53 cases treated with repeated SBRT, 13 cases occurred in the same segment, and 40 cases occurred in a remote segment.

Table 2 shows the background characteristics of patients with recurrence in the same segment, or a remote segment treated by repeat SBRT. We compared age, sex, etiology, Child–Pugh score, platelet count, albumin, total bilirubin, prothrombin time, ALBI score, tumor size, number of nodules, alpha-fetoprotein, des-γ-carboxy prothrombin, Barcelona Clinic Liver Cancer, and previous TACE. However, no significant differences were identified.

Figure 5 shows OS for the same-segment and remote-segment groups. Median OS for the same segment was 92 months (95% CI, 49 months–NA), and that for the remote segment was 70 months (95% CI, 41 months–102). No significant difference was evident (*p* = 0.53).

Figure 6 shows box-and-whisker plots for the transition of ALBI scores at 3, 6, and 12 months after irradiation. No significant difference in median ALBI score was seen between same- and remote-segment groups before irradiation. Likewise, no significant differences were apparent 3 months after irradiation. However, ALBI scores after 6 and 12 months were significantly different in the same-segment (*p* = 0.015) and remote-segment (*p* = 0.030) groups.

### 3.4. Treatment-Related Toxicities

Table 3 shows side effects in the total and repeated groups, and in the same-segment and remote-segment groups. The types of side effects were bile duct stenosis (two cases), portal thrombosis (one case), pneumonia (six cases), gastrointestinal toxicities (three cases), ascites (two cases), and dermatitis (one case). Only one case showed G3 portal vein thrombosis; that case achieved resolution with oral anticoagulant medication. G3 bile duct stricture was observed in the same segment of repeated SBRT. The patient had received previous surgery and TACE for HCC. SBRT was administered as 48 Gy in four fractions for recurrent HCC in segment 8. After 66 months, recurrent HCC was diagnosed in the same segment and reradiation was administered as 40 Gy in four fractions. Bile duct stricture was diagnosed 2 months later. The stenosis was at the root of the right hepatic duct and total maximum biliary dose at the stenosis of the bile duct from the first and second SBRT was 75.5 Gy. Endoscopic retrograde cholangiopancreatography was subsequently performed, and endoscopic biliary stenting was performed. The patient subsequently underwent four stent replacements over the next 2 years and eventually died of liver failure.

## 4. Discussion

Early-stage HCC is generally treated by RFA or operation according to the BCLC Staging System or JSH HCC Guidelines 2021, which were published by the Japan Society of Hepatology. The SURF trial conducted in Japan did not clearly identify whether surgery or RFA was the better treatment option for small HCC [19].

Although one report compared the advantages of hepatectomy and SBRT [20], the data remain inadequate to draw definitive conclusions. In a retrospective comparison of SBRT and RFA outcomes using propensity score analysis with liver function as an adjustment factor, SBRT was reported to show no difference in prognosis, but significantly better local control compared to RFA [21,22], suggesting that SBRT may offer comparable efficacy to RFA.

Several prospective studies have reported high local control and survival rates for SBRT in HCC [7,8,9,13]. Most of the prospective studies were conducted in patients who deemed refractory to surgery or RFA [7,8,13,23], suggesting that SBRT may represent an effective local treatment for patients with HCC who are refractory to surgery or RFA. In our report, only six cases experienced LR, of which three had multiple intrahepatic recurrences not only in the PTV margin, so both LR and accidental intrahepatic recurrence of a single lesion in the PTV may have occurred. Just three cases showed recurrence within the PTV alone, so SBRT was suggested to provide very good local control. Regarding OS, it is necessary to consider that patients eligible for SBRT may be unsuitable for resection or RFA, and thus may also be relatively old and frail. In addition, the fact that the majority of patients (78%) had received other treatments such as surgery, RFA, or TACE prior to SBRT may also have affected the results.

In general, treatment after recurrent HCC is the same as that for first-episode cases, including surgery and RFA.

In a report comparing treatment for recurrent HCC after hepatic resection, reresection was associated with better prognosis when comparing results between reresection and non-resection treated cases [24,25,26]. In addition, some studies have reported the equivalent results when comparing surgery and RFA for recurrent HCC after hepatic resection [27,28], while others have reported no significant difference in either OS or DFS when comparing surgery (including transplantation) and RFA for recurrence after RFA [29,30].

Rossi et al. reported 3- and 5-year survival rates of 67% and 40.1%, respectively, in a study of 696 patients who underwent repeated RFA for recurrent HCC after RFA [31].

Treatment of recurrent lesions after SBRT is limited, as reported by the high incidence of biliary leakage when liver resection is performed for recurrent HCC after radiotherapy [20]. Therefore, in the event of intrahepatic recurrence after SBRT, we must consider the need for repeated SBRT as local therapy. Kimura et al., reported that repeated SBRT for intrahepatic recurrent HCC resulted in high local control rates with good safety and satisfactory OS comparable with that of other curative local treatments for patients with well-preserved liver function [32]. Eighty-one patients treated with repeated SBRT were included in this study. Median OSs for first and second SBRT were 71 months (95% CI, 59.9–82.1 months) and 44 months (95% CI, 38.7–9.3 months), respectively. Only seven cases recurred locally.

When repeated SBRTs are performed, consideration needs to be paid to where the recurrent lesion is located in the liver. This is because, if the recurrent lesions to be irradiated are close to the previous lesion, the irradiation will be concentrated in one part of the liver, which may lead to a decrease in liver function and complications. We therefore divided recurrent lesions after SBRT into same-segment and remote-segment lesions. Moreover, evaluation of RILD after repeated SBRT is important. RILD is characterized by the fibrotic occlusion of small hepatic veins, resulting in congestion and hepatic cell loss, and is seen within 3 months after treatment [33]. RILD is typically characterized by hepatomegaly and nonmalignant ascites 2–6 weeks after the end of irradiation, as well as elevated ALP. The increase in the risk of RILD with repeated SBRT should also be considered. Lo et al. reported the safety of repeated SBRT in 14 patients with 18 lesions using CyberKnife. One Child–Pugh class A patient developed non-classic RILD, and three patients had Child–Pugh class progression after the second course of SBRT. Although this study had some limitations, absence of RILD after the first course of SBRT, longer interval between first and second SBRT, presence of Child–Pugh class A or B disease, low tumor burden, or good performance status may indicate suitability for repeat SBRT [34]. No cases of RILD were encountered in our study and no significant differences in liver function were seen during the first 6 months after repeated SBRT in the same or a remote segment. ALBI scores were significantly elevated 6–12 months after irradiation in both the same and remote segments. ALBI scores are sufficiently sensitive for detecting the early deterioration of liver function during the treatment of HCC [35], so this is thought to represent a purely sequential decrease in liver function rather than a side effect of irradiation. The decrease in liver function seen in cases of repeated SBRT was found to be independent of the irradiation site. Furthermore, no significant difference in OS was seen depending on the location of irradiation for the recurrent lesion.

As for side effects, neither G3 pneumonia nor gastrointestinal toxicities occurred, but G3 bile duct stricture was observed in only one case with irradiation in the same area. The cumulative dose at the point of bile duct stricture was 75.5 Gy. Eriguchi et al. reported that bile duct stricture attributed to SBRT occurred only in cases in which the total dose exceeded 80 Gy after repeated SBRT, and that Grade 2 or higher toxicity did not occur with normal single irradiation [36]. In our report, bile duct stricture occurred due to the effect of repeated SBRT to the same segment in a case close to the central biliary tract, although 66 months had passed since the initial irradiation. Since the total dose that can cause bile duct stricture is still unclear, irradiation for patients with recurrence may result in bile duct stricture in some cases where the irradiated areas are close to each other.

Although various dose protocols for HCC are used internationally, it is reported that the central dose of the biologic effective dose (BED) 10 Gy ≥ 100 Gy should be met [37]. In our hospital, dose protocol was standardized at 40 Gy in four fractions, and the central dose was 50 Gy, which met BED10 Gy ≥ 100 Gy. While the bile duct dose needed to be carefully considered, it was presumed that the same dose protocols as for the first SBRT was the best for repeated SBRT, since the indications for repeated SBRT were basically the same as for the first SBRT.

## 5. Conclusions

Repeated SBRT is an effective treatment for HCC, offering high local control rates and minimal toxicity. No RILD due to repeated SBRT was encountered and the decrease in liver function after repeated SBRT did not depend on the areas irradiated.

## Figures and Tables

**Figure 1 cancers-15-00846-f001:**
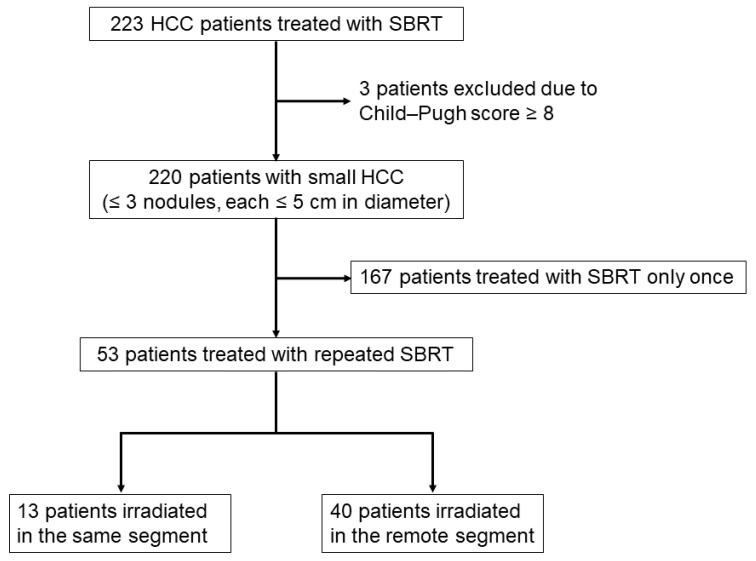
Flow chart of patient selection in the study.

**Figure 2 cancers-15-00846-f002:**
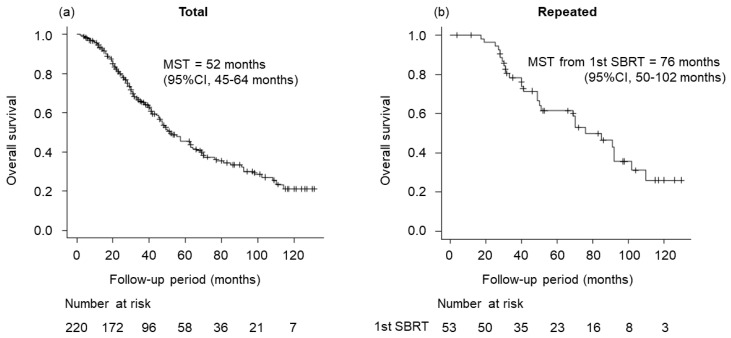
Overall survival (OS) for total cases treated with stereotactic body radiation therapy (SBRT) and repeated cases treated with SBRT. (**a**) OS for total cases (median survival time (MST), 52 months; 95% confidence interval (CI), 45–64 months). (**b**) OS for repeated cases. MST from first SBRT was 76 months (95% CI, 50–102 months).

**Figure 3 cancers-15-00846-f003:**
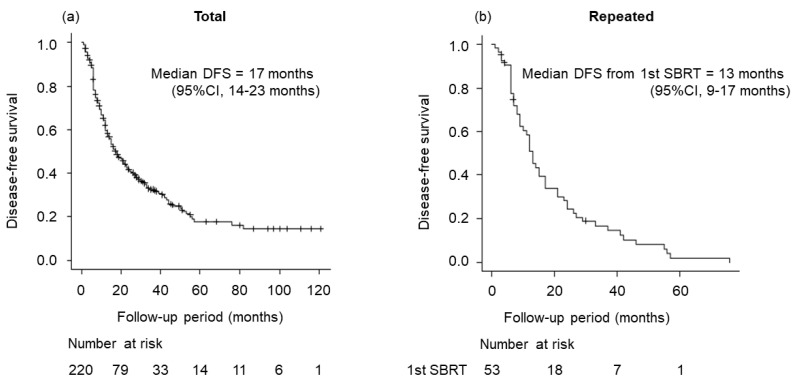
Disease-free survival (DFS) for total cases treated with SBRT and repeated cases treated with SBRT. (**a**) DFS for total cases (median DFS, 17 months; 95% CI, 14–23 months). (**b**) DFS for repeated cases (median DFS from first SBRT, 13 months; 95% CI, 9–17 months).

**Figure 4 cancers-15-00846-f004:**
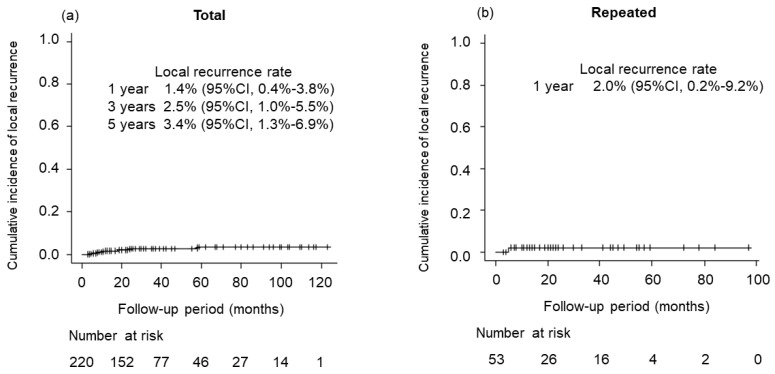
Local recurrence (LR) rate for total cases treated with SBRT and repeated cases treated with SBRT. (**a**) Three- and 5-year LR rates for total cases were 2.5% (95% CI, 0.4–3.8%) and 3.4% (95% CI, 1.3–6.9%), respectively. (**b**) Only one case showed LR within 1 year after second SBRT.

**Figure 5 cancers-15-00846-f005:**
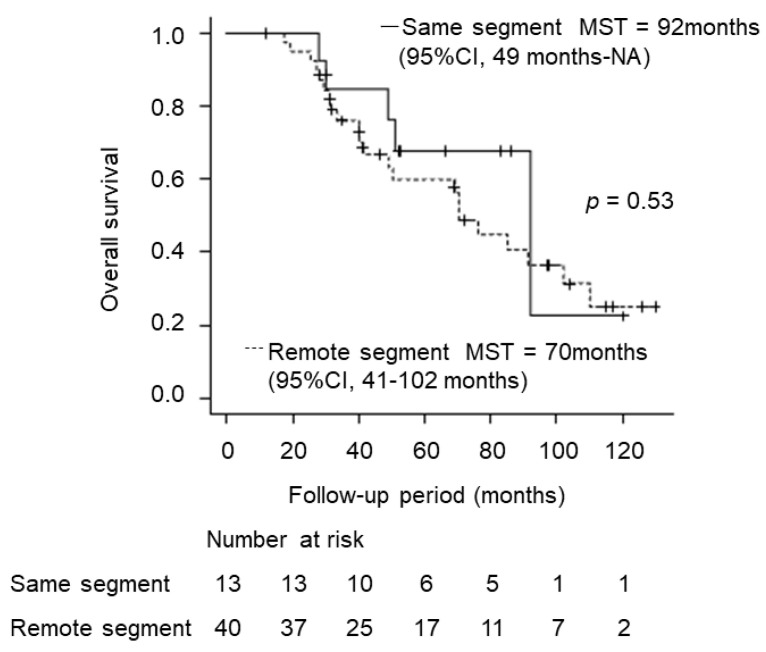
Overall survival (OS) for same-segment and remote-segment groups. Median OS for the same-segment group was 92 months (95% CI, 49 months–NA), and that for the remote-segment group was 70 months (95%CI, 41–102 months), showing no significant difference (*p* = 0.53).

**Figure 6 cancers-15-00846-f006:**
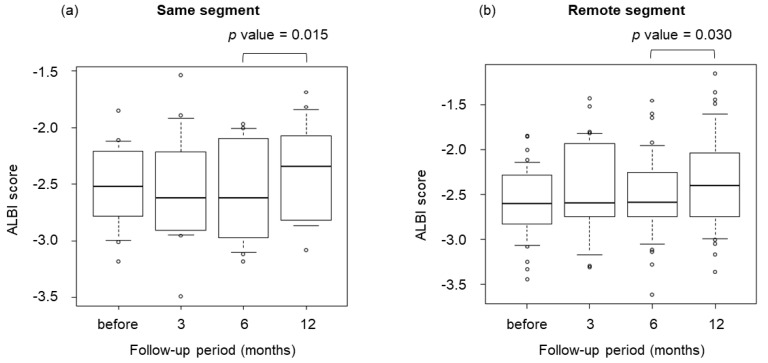
Box-and-whisker plots of albumin–bilirubin (ALBI) scores at 3, 6, and 12 months after SBRT. ALBI scores after 6 months and after 12 months were significantly different in both the same-segment group (*p* = 0.015) and remote-segment group (*p* = 0.030).

**Table 1 cancers-15-00846-t001:** Background characteristics for total cases (*n* = 220) and repeated cases (*n* = 53).

Characteristic	Total (*n* = 220)	Repeated (*n* = 53)
Age, range, *y*	76 (38–95)	76 (54–89)
Sex (male/female), *n*	143/77	40/13
Etiology (HBV/HCV/NBNC), *n*	32/143/48	7/39/7
Child–Pugh score (5/6/7), *n*	148/49/23	41/9/3
Platelet count, range, ×10^4^/μL	11.5 (2.8–33.2)	10.8 (4.6–32.7)
Albumin, range, g/dL	3.9 (2.7–5.1)	4.1 (3.0–5.0)
Total bilirubin, range, mg/dL	0.8 (0.2–2.8)	0.8 (0.4–2.0)
Prothrombin activity, range, %	85 (32–119)	87 (43–119)
ALBI score, range	−2.60 (−3.70–1.31)	−2.76 (−3.70–1.80)
Size of main tumor, range, mm	16 (8–50)	17 (8–35)
Number of tumors (1/2), *n*	201/19	50/3
Treatment-naïve before first SBRT/Recurrence, *n*	48/172	6/47
Serum AFP value, range, ng/mL	6.5 (0.5–4470)	8 (1.6–313.5)
Serum DCP value, range, mAU/mL	26 (1.9–3811)	24.5 (10–592)
BCLC stage (0/A), *n*	101/119	24/29
Previous TACE before SBRT (with/without), *n*	191/29	46/7
Dose/fractions (prescription)		
40 Gy/4–5 fractions, *n*	112	27
48 Gy/4 fractions, *n*	86	21
60 Gy/8 fractions, *n*	22	5

Values represent median (range) or number of patients. HBV—hepatitis B virus infection; HCV—hepatitis C virus infection; NBNC—non-B, non-C viral hepatitis; ALBI—albumin-bilirubin; AFP—alpha-fetoprotein; DCP—des-γ-carboxy prothrombin; BCLC—Barcelona Clinic Liver Cancer; TACE—transarterial chemoembolization; SBRT—stereotactic body radiation therapy.

**Table 2 cancers-15-00846-t002:** Patient background for same-segment (*n* = 13) and remote-segment (*n* = 40) groups.

Characteristic	Same Segment (*n* = 13)	Remote Segment (*n* = 40)	*p* Value
Age, range, y	78 (58–91)	78.5 (54–92)	0.528
Sex (male/female), *n*	12/1	28/12	0.148
Etiology (virus/non virus), *n*	12/1	36/4	1
Child–Pugh score (5/6,7), *n*	9/4	27/13	1
Platelet count, range, ×10^4^/μL	12.1 (5.8–24.8)	12.1 (4.7–24)	0.188
Albumin, range, g/dL	3.8 (2.8–5.1)	3.9 (3.1–4.9)	0.305
Total bilirubin, range, mg/dL	0.9 (0.4–1.4)	0.9 (0.3–1.6)	0.288
Prothrombin activity, range, %	96 (61–108)	91 (51–128)	0.23
ALBI score, range	−2.52 (−3.18–1.85)	−2.65 (−3.44–1.85)	0.441
Size of main tumor, range, mm	15 (10–22)	13 (5–50)	0.748
Number of tumors (1/2), *n*	12/1	33/7	0.662
Serum AFP value, range, ng/mL	3.6 (2.2–136.9)	5.3 (0.9–3255)	0.482
Serum DCP value, range, mAU/mL	26.5 (14–366)	24 (10–405)	0.549
BCLC stage (0/A), *n*	9/4	18/22	0.4
Previous TACE before SBRT(with/without), *n*	11/2	35/5	1

Values represent median (range) or number of patients. ALBI—albumin–bilirubin; AFP—alpha-fetoprotein; DCP—des-γ-carboxy prothrombin; BCLC—Barcelona Clinic Liver Cancer; TACE—transarterial chemoembolization; SBRT—stereotactic body radiation therapy.

**Table 3 cancers-15-00846-t003:** Grade 3 side effects for total cases (*n* = 288) and repeated cases (*n* = 53), including the same-segment group (*n* = 13) and remote-segment group (*n* = 40).

Side Effects	Total (*n* = 288)		Repeated SBRT All (*n* = 53)		Same Segment (*n* = 13)		Remote Segment (*n* = 40)	
	Any grade	Grade3	Any grade	Grade 3	Any grade	Grade 3	Any grade	Grade 3
Bile duct stenosis	2 (0.69%)	1 (0.35%)	1 (1.9%)	1 (1.9%)	1 (7.7%)	1 (7.7%)	0	0
Portal thrombosis	1 (0.35%)	1 (0.35%)	0	0	0	0	0	0
Pneumonia	6 (2.1%)	0	2	0	0	0	2 (5.0%)	0
Gastrointestinal toxicities	3 (1.0%)	0	0	0	0	0	0	0
Ascites	2 (0.69%)	0	0	0	0	0	0	0
Dermatitis	1 (0.35%)	0	0	0	0	0	0	0

## Data Availability

The data that support the findings of this study are available from the corresponding author upon reasonable request.

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
