# Peer review of "Effectiveness of Repeated Stereotactic Body Radiation Therapy for Hepatocellular Carcinoma—Consideration of the Locations of Target Lesions"

_cancers, 2023, doi:10.3390/cancers15030846_

Round 1

Reviewer 1 Report

The authors analyzed the usefulness and safety of SBRT for hepatocellular carcinoma and found that repeated SBRT has high local control and low risk especially. The number of cases is sufficient, and the paper addresses an unmet need. We believe that this paper is useful mainly for specialists involved in the treatment of hepatocellular carcinoma.

Minor points

1.What is the mechanism of portal thrombosis, a Grade 3 adverse event?

2.The cases with different irradiation methods of SBRT are included. Did the irradiation       method make any difference in the results?

Author Response

1.What is the mechanism of portal thrombosis, a Grade 3 adverse event?

Irradiation causes vascular endothelial damage, which results in portal vein thrombosis.

It is reported that D2 of the portal vein of 40 Gy or more was a common point of portal vein thrombosis 1.

In our case, the site of portal vein thrombosis requiring medical treatment was 80% within the isodose, and the effect of SBRT could not be ruled out, so it was considered a Grade 3 adverse event.

  1. Takahashi, S.; Kimura, T.; Kenjo, M.; Nishibuchi, I.; Takahashi, I.; Takeuchi, Y.; Doi, Y.; Kaneyasu, Y.; Murakami, Y.; Honda, Y.; et.al. Case reports of portal vein thrombosis and bile duct stenosis after stereotactic body radiation therapy for hepatocellular carcinoma. Hepatol Res 2014, 44 (10), E273-8.

2.The cases with different irradiation methods of SBRT are included. Did the irradiation method make any difference in the results?

➝There were no differences between irradiation methods.

Reviewer 2 Report

In this study, the authors analyzed Effectiveness of repeated stereotactic body radiation therapy for hepatocellular carcinoma. This manuscript retrospectively evaluated the efficacy of stereotactic body radiation therapy, including repeated radiation therapy, for hepatocellular carcinoma. However, there are problems with the manuscript that need to be addressed.

1.Did these patients receive systemic therapy such as immunotherapy or target therapy especially in repeated SBRT group? How do you balance this factor because the effect of systemic therapy would have a great influence on DFS and OS.

2. How many people underwent metastasis beyond liver? This is an important factor of OS. You should analyze the value of repeated SBRT in a more comprehensive way.

3. It seemed repeated SBRT could improve OS for HCC patients(72m vs 52m), and Three- and 5-year OS rates for repeated cases were better either. Why the author did not analyze it with statistical methods? and why the author did not mention the OS benefit of repeated SBRT in conclusion. 

4. What is the value of ALBI in this study? 

5. Why did the author analyze the G3 side effects only? I suggest the author should record the rate of all Treatment-related toxicities. 

6. What is the best protocol in first SBRT plan? and what is the best protocol in repeated SBRT plan? The author should give a suggestion and specify the reason in conclusion. 

Author Response

1. Did these patients receive systemic therapy such as immunotherapy or target therapy especially in repeated SBRT group? How do you balance this factor because the effect of systemic therapy would have a great influence on DFS and OS.

Systemic therapy was administered before and after in 24 of 167 single SBRT cases and 9 of 53 repeated SBRT cases. On the other hand, only 27 cases of single SBRT and 5 cases of repeated SBRT were treated with SBRT alone for HCC, and other treatments were added in other cases.

Most of cases were performed surgery, TACE, or systemic therapy before or after SBRT, and as you pointed out, they certainly affect OS, so it must be evaluated only after SBRT. However, DFS is not affected by other treatments. DFS is helpful in evaluating the effectiveness of SBRT-only treatment.

2. How many people underwent metastasis beyond liver? This is an important factor of OS. You should analyze the value of repeated SBRT in a more comprehensive way.

➝There were no cases who underwent metastasis beyond liver at the time of radiotherapy.

3. It seemed repeated SBRT could improve OS for HCC patients(72m vs 52m), and Three- and 5-year OS rates for repeated cases were better either. Why the author did not analyze it with statistical methods? and why the author did not mention the OS benefit of repeated SBRT in conclusion. 

The repeated cases are considered to have an inevitably better OS because the patients maintained their liver function after the initial SBRT and the recurrent lesions had tumor factors that could be irradiated (1-2 tumors, ≤5 cm in size), and therefore, the prognosis is not considered to be improved by repeated SBRT.

4. What is the value of ALBI in this study? 

The ALBI grade model was used because it is a continuous variable and is suitable for comparing between values before and after treatment.

5. Why did the author analyze the G3 side effects only? I suggest the author should record the rate of all Treatment-related toxicities. 

➝I am really sorry. I have added it in the article.

6. What is the best protocol in first SBRT plan? and what is the best protocol in repeated SBRT plan? The author should give a suggestion and specify the reason in conclusion. 

➝I am really sorry. I have added the sentences in the last of discussion.

Although various dose protocols for HCC are used internationally, it is reported that the central dose of the biologic effective dose (BED)10 Gy≥100 Gy should be met. In our hospital, dose protocol was standardized at 40 Gy in 4 fractions, and the central dose was 50 Gy, which met BED10 Gy≥100 Gy. While the bile duct dose needed to be carefully considered, it was presumed that the same dose protocols as for the first SBRT was the best for repeated SBRT, since the indications for repeated SBRT were basically the same as for the first SBRT.

Reviewer 3 Report

Studies of re-SBRT are rare.

I think this is a good study because that Studies of re-SBRT are rare and that provides a basis for inspiring re-SBRT.

2.2 Therapeutic method

Please describe in detail how the CT was taken.

What dose constraints did you have for OAR (especially liver)?

Did you ever make any prescription reductions at the time of re-irradiation, such as dose constraints for the liver?

Author Response

1. Please describe in detail how the CT was taken.

Since there were cases of early enhancement effect that persisted for more than 3 months after SBRT, it is premature to determine efficacy early after treatment, and it seems appropriate to determine efficacy at about 6 months. In this study, patients underwent dynamic CT every 3, 6, and 12 months after completing SBRT.

2. What dose constraints did you have for OAR(organ at risk) (especially liver)?

As for intrahepatic organs at risk, there are reports of cases of bile duct stenosis after multiple SBRT of 80 Gy or more, but single SBRT did not cause bile duct stenosis of G2 or more. With regard to portal vein thrombosis, we must be careful when D2% of portal vein is more than 40 Gy. For a single SBRT of 40 Gy/4 fr, dose constraints to the portal vein and bile duct did not need to be considered, but when repeated SBRT were performed, these were taken into consideration.

3. Did you ever make any prescription reductions at the time of re-irradiation, such as dose constraints for the liver?

Repeated SBRT was performed using the same protocol as for the first SBRT, with no dose reduction.

Round 2

Reviewer 2 Report

The authors have made a full response to my qustions. "There were no metastasis occurred beyond the liver", I suggest the author should mention this in the manuscript.